# Gender Differences in Insulin Resistance: New Knowledge and Perspectives

Tiziana Ciarambino [1,*,†] , Pietro Crispino [2,†], Gloria Guarisco [3] and Mauro Giordano [4]

1   Internal Medicine Department, Hospital of Marcianise, 81100 Caserta, Italy
2   Internal Medicine Department, Hospital of Latina, 04100 Latina, Italy; pcr@libero.it
3   Diabetology, University Sapienza of Rome, Hospital of Latina, 04100 Latina, Italy; ggua@libero.it
4   Internal Medicine Department, University of Campania, L. Vanvitelli, 81100 Naples, Italy; mgio@libero.it
*   Correspondence: tiziana.ciarambino@gmail.com
†   These authors contributed equally to this work.

**Abstract:** Insulin resistance is the main mechanism in a whole series of pathological conditions, which are not only of metabolic interest but also of a systemic type. This phenomenon means that the body's cells become less sensitive to the hormone insulin, leading to higher levels of insulin in the blood. Insulin resistance is a phenomenon that can be found in both men and women and in particular, in the latter, it is found mainly after menopause. Premenopause, hormonal fluctuations during the menstrual cycle, and the presence of estrogen can affect insulin sensitivity. Androgens, such as testosterone, are typically higher in men and can contribute to insulin resistance. In both sexes, different human body types affect the distribution and location of body fat, also influencing the development of diabetes and cardiovascular disease. Insulin resistance is also associated with some neurological and neurogenerative disorders, polycystic ovary syndrome, atherosclerosis, and some of the main neoplastic pathologies. A healthy lifestyle, including regular physical activity, a balanced diet, and self-maintenance, can help to prevent the onset of insulin resistance, regardless of gender, although the different habits between men and women greatly affect the implementation of preventative guidelines that help in fighting the manifestations of this metabolic disorder. This review may help to shed light on gender differences in metabolic diseases by placing a necessary focus on personalized medical management and by inspiring differentiated therapeutic approaches.

**Keywords:** gender; insulin resistance; hormones





## 1. Background

Gender plays an important role in determining metabolism, both in normal subjects and in subjects affected by dysfunction related to insulin resistance (IR). Studying these relationships may provide insight into how hormonal changes during different life stages, such as puberty, pregnancy, and the menopause, affect IR and diabetes risk. Expanding our knowledge on the role of gender in IR makes it possible to identify specific risk factors and more effective preventative strategies [1–6]. In particular, women have an increased risk of IR during menopause; this information can be used to design targeted interventions to reduce this risk. Additionally, as regards their effectiveness, treatments can be diversified for both sexes, and this can provide more detailed information on therapeutic options and on the dosages of individual drugs [6]. To this day, it is difficult to find clinical guidelines that consider specific factors related to gender in their practice; therefore, a critical review of the available literature could help to elaborate upon each aspect related to IR, a serious infliction, with the aim of improving health outcomes for this condition.

## 2. Methods

We conducted a systematic literature search in PubMed to identify all types of studies published up to 31 May 2023 that examined the association between gender and insulin

resistance. We used the following search: ('Gender' OR 'Gender medicine') AND ('insulin resistance' OR 'metabolic syndrome' OR 'type 2 diabetes' OR 'steatohepatitis'). The inclusion criteria were as follows: clinical, molecular, experimental, and observational studies and meta-analyses. The exclusion criteria were as follows: comments, animal studies, or case reports, or studies that were duplicated or repeated or showed a great similarity in their sample or content to another study. The data extraction and quality assessment were conducted by two reviewers (PC and TC), who independently extracted the following data and assessed the quality of each study. Any discrepancies between the reviewers in the research selection, quality assessment, or data extraction were addressed via re-evaluating the original with two other authors (GG and MG). The supervision of all work carried out was conducted by MG, the creator and coordinator of the research.

## 3. Prevalence

IR is the inability of exogenous or endogenous insulin to perform its role in glucose uptake and utilization in a normal population [1]. IR is part of a more complex syndrome characterized by metabolic dysfunction and an increased cardiovascular risk, defined as metabolic syndrome (MS), IR syndrome, or syndrome x, a cluster of medical conditions and risk factors that increase the risk of developing heart disease, stroke, and type 2 diabetes (T2DM). Due to the absence of specific parameters, it is difficult to define and identify insulin resistance and to study its prevalence, but the International Diabetes Federation (IDF) estimates that around 25% of the world's population has MS [2]. Other epidemiological studies have estimated that the worldwide prevalence of MS is between 10% and 84%, depending on the ethnicity, age, sex, and race of the population, the total adiposity, and fat distribution [3,4]. The results of the National Health and Nutrition Examination survey showed that the prevalence of MS is around 5% in normal-weight people, 22% in overweight people, and 60% among obese people [1].

## 4. Body Fat Distribution

The body composition and body fat amount and distribution differ among males and females. Many studies demonstrate sexual dimorphism in the human gene expression responsible for regional fat distribution [5]. In fact, during their reproductive life, women are characterized not only by a higher total fat mass and a reduced lean mass compared to men, but also by a different site of adipose tissue deposition compared to men. Women are characterized by a "gynoid fat distribution", which consists of the tendency to store adipose tissue mainly in the subcutaneous sites and especially in the gluteal–femoral regions. Furthermore, women are less susceptible than men to the depositing of fat at ectopic sites, such as in the liver or pancreas, or in the epicardium. On the contrary, men are more likely to accumulate fat in the visceral areas of the trunk/abdominal region, determining the typical "android fat distribution" [5,6]. It is well known that ectopic adipose tissue is closely related to the onset of IR and metabolic diseases. Specifically, visceral adipose tissue (VAT) located in the main organs involved in glucose homeostasis, such as the liver or pancreas, may represent the trigger for IR onset and an important risk factor for T2DM and cardiovascular diseases (CVDs) [7,8]. For this reason, thanks to their different fat-storage capacities, women have a lower predisposition to T2DM than men, despite the higher prevalence of obesity among women globally [9]. This sex-specific fat distribution depends on the fat-storage capacity of the adipocytes of the subcutaneous adipose tissue and is regulated by genetic and hormonal factors [10]. A study suggests that sex steroid hormones contribute to sexual dimorphism in human body fat distribution and, in particular, underlines the protective role of endogenous estrogens [6], as explained below. However, sex steroids are not the only factors responsible for sexual dimorphism in body composition; there is evidence that genetic factors related to the sex chromosomes and epigenetic regulations influence weight gain, adiposity distribution, and glucose metabolism [11]. The main differences in body composition and metabolic features are summarized in Table 1.

**Table 1.** Principal differences in fat and glucose metabolism in female and male individuals.

| Female Metabolic Characteristics | Male Metabolic Characteristics |
| --- | --- |
| Increase in total fat mass | Prevalent skeletal muscle mass |
| Prevalent subcutaneous adiposity | Prevalent visceral adiposity |
| Increase in insulin sensitivity with age | Increase in ectopic fat |
| Prevalent NEFA storage at rest | Prevalent NEFA oxidation at rest |
| Prevalent NEFA oxidation during exercise | Prevalent glucose oxidation during exercise |
| Prevalent glycemia alteration after a meal | Prevalent glycemia alteration at fast |

## 5. Hormonal Factors

As mentioned, hormonal factors play an important role in gender-related differences related not only to the anatomical distribution of adipose tissue but also to many aspects of energy and glucose homeostasis. For this reason, the susceptibility to many diseases, including diabetes and metabolic disorders, is different between males and females. For example, thanks to the tendency to store adipose tissue in the subcutaneous tissue, preventing the accumulation of visceral and ectopic fat, insulin sensitivity and secretion are greater in women than in men. The protective role of endogenous estrogens in women is demonstrated by the negative consequences of menopause on body composition and on glucose homeostasis; in fact, menopause can trigger the progressive accumulation of visceral fat, leading to an increased incidence of metabolic disorders [12]. Furthermore, rare loss-of-function mutations in the gene encoding either aromatase (the enzyme that converts androgens into estrogens) or estrogen receptor $\alpha$ (ER$\alpha$) result in dysmetabolic phenotypes in individuals of both sexes [6]. The protective action of endogenous estrogens occurs through the activation of alpha receptors (ER$\alpha$) present in various tissues by improving and/or modulating glucose homeostasis [12]. Through their action on pancreatic beta-cell receptors, estrogens improve fasting insulin and glucose-stimulated insulin secretion and protect beta-cell function from apoptosis induced by metabolic injuries, such as oxidative stress and lipotoxicity [6]. In the liver, estrogens modulate gluconeogenesis, improve hepatic insulin response, and reduce hepatic insulin degradation [12]. As regards estrogen's action on fat and lean mass, this hormone improves insulin sensitivity in adipocytes and reduces adipocyte oxidative stress, while improving insulin-stimulated glucose uptake by skeletal muscle. ER$\alpha$ is also present in cardiac tissue, where it mediates the improvement of cardiac function and mitigates insulin resistance-induced cardiomyopathy. Finally, in the vascular endothelium, estrogens enhance nitric oxide production and increase vasodilation [12]. In light of this evidence, estrogen deficiency, as occurs during menopause, is a potential risk factor for the development of IR probably due to the reduction in circulating estrogens. In support of this hypothesis, it has been shown that surgically induced menopause increases the risk of developing IR and MS. Meanwhile, estrogen replacement therapy significantly improves insulin sensitivity and reduces the new onset of diabetes [12,13]. The effects of testosterone are mediated not only by the direct activation of the androgen receptors but also indirectly, through its aromatization into estrogens by the aromatase expressed by some tissues, such as the brain or the adipose tissue, well-recognized sites of estrogen biosynthesis [6]. In men with hypogonadism, low testosterone plasma concentrations are correlated with an increased risk of TD2M and vascular diseases, while testosterone supplementation improves glucose and lipid homeostasis [6]. Several works showed the role of testosterone in adipose tissue IR. Li X et al. demonstrated that testosterone has a particular role in the IR of adipose tissue according to its concentration and gender: in men, testosterone promotes insulin sensitivity, while in overweight or obese women it promotes IR of adipose tissue and, consequently, low testosterone concentrations in males and excess testosterone in females both contribute to adipose IR [14]. However, beyond absolute sex hormone concentration, an unbalanced androgen/estrogen ratio is associated with metabolic risks [14]. These considerations

underline the importance of including gender in the assessment of MS and cardiovascular risk to improve the management and prevention of these conditions. The protective effects of estrogens in various organs or tissues are summarized in Table 2.

**Table 2.** Protective effect of estrogens in the different specific organs and tissues.

| Organ/Tissue | Estrogen Activity |
|---|---|
| Adipose tissue | • Improves insulin sensitivity<br>• Reduces oxidative stress |
| Heart | • Prevents insulin resistance-induced cardiomyopathy |
| Liver | • Modulates gluconeogenesis<br>• Improves hepatic insulin response<br>• Regulates hepatic sensitivity in adipocytes |
| Pancreas | • Reduces insulin-releasing during fast<br>• Regulates insulin-secretion after glucose stimulation |
| Muscular system | • Improves insulin-stimulated glucose absorption |
| Vascular endothelium | • Increases nitric oxide production<br>• Improves vasodilatation |

## 6. Insulin Signal Transduction Perturbations in Insulin Resistance

The alterations of the mechanisms concerning the function of the insulin receptor and the transduction of its signal are the basis of the manifestations of IR [15–17].

The insulin receptor is a transmembrane glycoprotein consisting of two α and two β subunits linked by a pair bond via a disulfide bridge. After binding to insulin, the receptor undergoes phosphorylation which is also reflected in the intracellular substrates of the receptor itself [18,19]. The substrates that promote different signal transduction pathways are dependent on phosphoinositide-3-kinase (PI3K) which in turn phosphorylates the serine/threonine residue of protein kinase B (Akt) or the CAP/Cbl/TC10 pathway. Akt regulates the translocation of the glucose transporter GLUT4 to the cell surface via a phosphorylation of GTPase-activating protein. Akt also promotes glycogen synthesis through inhibition of GSK3 activity and induces protein synthesis through the activation of mTOR and downstream elements. Akt directly phosphorylates and inhibits the transcription factor FoxO, which inhibits autophagy. Hyperglycemia-induced AGE production and binding to their receptor RAGE impair insulin signal transduction by triggering a number of signaling pathways, including JNK, NF-κB, and PKC activation. Prolonged accumulation of AGEs depletes the expression of the cell-surface anti-AGE receptor AGER1, which is responsible for the inhibition of the deleterious effects of AGEs by competitively interfering with their binding to RAGE. AGER1 together with Sirtuin1 promotes the phosphorylation and activation of AMPK, which induces GLUT4 gene expression through the activation of MEFs and GEF transcription factors [15–20]. In normal subjects, the complex between insulin and its receptor activates a signal transduction process promoting phosphorylation of the enzyme phosphatidylinositol 3-kinase (PI3K), which leads to the formation of Akt 1 and 2 protein kinase C (PKC) which have the following metabolic effects [15,16]:

- Promotion of glucose entry into metabolism target cells by activating glucose transporter 4 (GLUT4) on the cell membrane in the liver, muscle, and adipose tissue, stimulating hepatic glycogen synthesis and inhibiting gluconeogenesis and glycogenolysis;
- Inhibition of lipolysis and stimulation of the synthesis of triglycerides, favoring their deposition in adipocytes (lipogenesis);
- Stimulation of protein synthesis by transcription and translation of mRNA in various cells;

- The phosphokinase pathway mediates the release of nitric oxide from endothelial cells (eNO), promoting vasodilation;
- Promotion of proliferation, contraction, and proinflammatory activity of smooth muscle cells.

Mechanisms involved in IR regard β-cell function and mass, insulin receptor substrate, phosphatidylinositol 3-kinase activity, protein kinase B/Akt, GLUT4 activity, mammalian target of rapamycin/mTOR, and AMP-activated protein kinase activity [15] (Figure 1). It has been recognized that insulin secretion and insulin sensitivity are regulated by pancreatic β-cells in a very definite manner to maintain homeostatic concentrations of plasma glucose in healthy individuals. During normal physiological conditions, there is a positive feedback loop between the β-cells and insulin-sensitive tissues [21,22]. The exhaustion of β-cells has a determinant role in the pathogenesis of the condition because it determines oxidative stress [23,24]. Initially, glucose along with elevated levels of free fatty acids (FFAs) leads to an increase in β-cell mass as a compensatory mechanism against IR [25]. The deposition of islet amyloid polypeptide (IAPP) or amylin in pancreatic islets is the other pathological mechanism involved in β-cell dysfunction. IAPP deposition causes a progressive decline in pancreatic islet cell mass, predominantly β-cells. Recently it was identified that the increase in expression of the receptor for advanced glycation end products (RAGEs) contributes to IAPP-induced β-cell proteotoxicity with oxidative stress, inflammation, and apoptosis β-cell pathogenesis, leading to islet inflammation and β-cell apoptosis [24,26,27]. The insulin receptor is a key mediator of insulin action [28]. Insulin receptor 1 (IRS1) and insulin receptor 2 (IRS2) differ significantly in their abundance and tissue-specific function. A significantly lower expression of IRS1 was observed in morbidly obese and insulin-resistant subjects [29]. In insulin resistance, IRS1 malfunctions more often than IRS2 [28]. PI3K is a key regulatory node in the translation of insulin-activated extracellular signals [30]. PI3K consists of an 85 kDa regulatory subunit and a 110 kDa catalytic subunit. The former is responsible for the negative regulation of PI3K [31]. IR occurs when there is an inhibition of PIK3 due to a mutation involving PIK3R1 which regulates the correct functioning of the regulatory subunit [32]. The serine/threonine protein kinase Akt has three isoforms but Akt2 is the most important one in insulin-mediated glucose uptake and lipid metabolism [33]. Akt plays a crucial role in cell survival and growth through its action on insulin-stimulated glucose uptake, glycogen, and protein synthesis. The depletion of Akt or its inhibition is associated with hyperglycemia and insulin resistance [34]. Any defect in the PI3K/Akt/AS160 signaling will eventually reduce the glucose uptake in insulin-sensitive tissues and lead to insulin resistance [35]. Glucose transporter proteins (GLUTs) are responsible for the transport of glucose across the plasma membrane. In particular, GLUT4 is insensitive to insulin and is responsible for glucose homeostasis [36]. Depletion of GLUT4 expression has been shown to be associated with IR [37]. GLUT4 gene expression is under the dynamic control of specific transcription factors, whereby any deletion or truncation of specific sequences leads to suppression of GLUT4 mRNA expression [37]. Chronic activation of mTOR, as in the case of overnutrition or obesity, leads to insulin resistance through a negative feedback loop through IRS degradation [38]. mTOR has a crucial role in the growth and survival of β-cells and insulin secretion [39]. The mTOR pathway plays a key role in the regulation of cellular growth and proliferation [38]. Activated protein kinase AMPK is a central regulator of multiple metabolic pathways and is activated by a diminished cellular energy state. AMPK plays a substantial role in glucose uptake by upregulating GLUT4 expression [15]. AMPK is generally correlated with the increase in insulin sensitivity in the body and a decrease in insulin resistance as it is the inhibitor of acute proinflammatory responses and protects against obesity-induced insulin resistance [40].

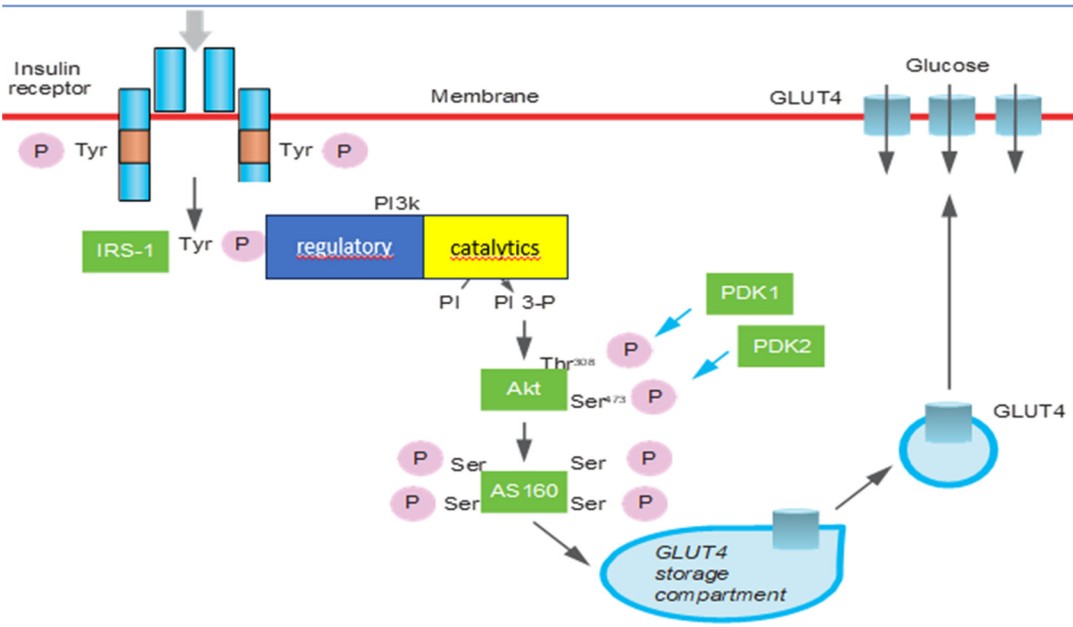

**Figure 1.** The mechanisms that regulate the translocation of glucose transporters on the plasma membrane, the synthesis of glycogen, and the biosynthesis of GLUTs in skeletal muscle and adipose cells. The signal is transmitted through the sequential activation of the insulin receptor and of the IRS-1 proteins, PI 3-kinase, Akt, and AS160. The kinase PDK1 and the hypothetical kinase PDK2 contribute to the activation of Akt, responsible for the phosphorylation of Akt at the level of threonine 308 and serine, respectively.

### 7. Associated Health Conditions

IR manifests itself from the clinical point of view in various ways, involving various organs and systems, including metabolic syndrome and TD2M, nonalcoholic fatty liver disease (NAFLD), the syndrome of polycystic ovary disease (PCOS), obstructive sleep apnea syndrome (OSAS), atherosclerosis, cardiovascular disease, some neurodegenerative diseases, and neoplastic diseases (Table 3). This polyvalence in determining rather different pathologies is attributable to the fact that insulin exerts physiological effects on lipid and protein metabolism and participates with its anabolic function in the division and proliferation of the cell without neglecting the fact that its receptors are distributed in various organs and tissues of the body [41]. Although IR is typically a male condition, the study of gender-specific differences has become important since the protective role of estrogens, which reduce the risk of this disorder among women, was discovered [42–44].

**Table 3.** Associated health conditions of insulin resistance.

| Associated Health Conditions of Insulin Resistance |
|:---:|
| Metabolic syndrome |
| Type-2 diabetes mellitus |
| Non-alcoholic fatty liver disease |
| Syndrome of polycystic ovary disease |
| Atherosclerosis and cardiovascular disease |
| Neurodegenerative disease(Parkinson's disease and Alzheimer's disease) |
| Neoplastic diseases(breast, colorectal, pancreas, and liver cancer) |

### 7.1. Metabolic Syndrome

MS is classically associated with obesity and insulin resistance contributing to a series of cardiometabolic alterations that result in increased cardiovascular risk due to the fact that visceral adipose tissue produces a greater amount of proinflammatory adipokines that induce a permanent inflammatory state in this tissue by reducing its sensitivity to insulin [45–47]. Plasma glucose and triglycerides are diverted to other organs (pancreas, liver, kidney, blood vessels, skeletal muscle, heart, and epicardial adipose tissue, EAT), altering their insulin sensitivity [45–47]. Proinflammatory cytokines produced by the adipocytes play a crucial role in the development of fatty liver disease. In fact, they reduce hepatocyte insulin sensitivity by favoring the deposition of lipids in the liver, fibrosis, and steatosis (nonalcoholic fatty liver disease, NAFLD), which in turn worsens the state of insulin resistance [48]. Metabolic syndrome (MS) is a critical factor related to NAFLD, to the extent that NAFLD has been considered by many to be the hepatic manifestation of this syndrome [49]. A higher prevalence is known of NAFLD in men, associated with visceral obesity, IR, and dyslipidemia [50]. Although NAFLD is a condition more frequent in males with IR and dyslipidemia, we found that there is a gender-specific association between these variables and the female population [51–54]. Women tend to have more subcutaneous adipose tissue, higher basal leptin levels, and elevated estrogen levels, while less thick adipose tissue in the visceral compartment. On the contrary, males have higher peripheral IR and a higher content of fatty acids at the level of the portal and enterohepatic circulation and this favors the development of metabolic syndrome, NAFLD, and cardiovascular consequences [55–59]. A particular role in the evolution of MS has been recognized for androgens. Studies show that excess female androgens and male androgen deficiency have similar metabolic phenotypes, showing the complexity of androgens' role in metabolism [60–63]. In NAFLD, females show greater IR, hypertriglyceridemia, and visceral adiposity, while in males it is conceivable that there are factors that go beyond IR. In particular, we focused on the lipid metabolism in the lipolytic capacity of both sexes. These studies demonstrate that females have more lipolysis while males show prolonged de novo lipogenesis, which may be associated with the damage to lipid storage typical of NAFLD [64]. Beta-oxidation during fasting in men is also less efficient than in females [64]. It would seem that such metabolic disturbances occur until attenuated with the exogenous administration of androgens and estrogens [65–67]. Srinivas et al. [65] investigated, from a gender perspective, the association of NAFLD with the risk factors of the components of MS and found that the components related to this disorder in women were hyperglycemia and hypertriglyceridemia, while in men only the BMI was related. In another study, the risk component among males was the indirect measure of adiposity (WC), while in females it was low HDL levels and hypertriglyceridemia [66]. From a practical point of view, the presence of MS seen from a gender perspective cannot be diagnosed only with the measurement of fasting blood sugar but must be correlated with a more in-depth metabolic study [67]. The prevalence of glucose metabolism disorder differs between genders, giving rise to different clinical implications: men more often develop abnormal fasting glucose levels, while women more often show elevated glucose levels after meals. Fasting glucose elevation is characterized by increased hepatic glucose production and decreased early insulin secretion, whereas postprandial hyperglycemia is mainly due to peripheral IR [68]. Postprandial hyperglycemia carries a higher risk of mortality since it is more strictly a cause of increased cardiovascular risk, therefore, if initial MS is suspected, oral glucose tolerance tests should be performed to screen, especially in women [69,70]. As we have already mentioned, women enjoy particular protection against MS from estrogen and this lasts until menopause unless there is a hypoestrogenic state or a prolonged anovulatory condition [69]. Together with estrogen, vitamin D can also directly stimulate the expression of the insulin receptor, thus improving the transport of glucose into human cells [70]. A significant interaction between sex and vitamin D is evidenced by the fact that low levels of 25(OH) of vitamin D3 are associated with T2DM in women but not in men [71]. We have also seen how there is an interchange between vitamin D and estrogen; on the one hand,

vitamin D increases the bioavailability of estrogens and, on the other, the latter is able to increase the efficiency of absorption, transport systems, and affinity with its receptor [72]. The important role of the composition of the intestinal microbiota has also been established both in promoting the production of intestinal estrogens with a systemic function and, with them, greater absorption of vitamin D [73]. The effect of estrogen deficiency in menopausal women is associated with an increased risk of T2DM and manifests itself through three different mechanisms including impaired insulin secretion by pancreatic beta cells, reduced sensitivity to insulin by the target organs and tissues, and increased sensitivity to glucose by the main organs involved in diabetes-related pathology [74,75].

### 7.2. Atherogenesis and Endothelial Damage

We observed that insulin resistance is related to an alteration of the lipid characterized by hypertriglyceridemia, increased concentration of very low-density lipoprotein (VLDL), decreased concentration of high-density lipoprotein (HDL), and formation of low-density lipoprotein (LDL) [76–78]. Insulin regulates lipid metabolism through the PI3K pathway, promoting the degradation of apoprotein B and hindering the production of VLDL by the liver. In the presence of insulin resistance, on the contrary, there is both increased production and reduced clearance of VLDL [79]. The hepatic protein CEPT instead intervenes in promoting the transfer of triglycerides from VLDL to LDL and HDL, increasing the overall triglyceride content of these particles [80,81]. Triglyceride-rich LDL and HDL become the substrate of liver lipase, which promotes the removal of triglycerides from these lipoproteins, thereby transforming LDL into sdLDL and reducing apoprotein A (apo-A) concentration, increasing the catabolism of HDL [80,81]. This transfer process of fatty acids is strongly atherogenic since sdLDL fills the vascular wall more diffusely than LDL, has a longer half-life, is more oxidizable than LDL macroparticles, and has a lower affinity for the LDL receptor [41]. It is therefore clear that IR intervenes synergistically with vascular blood flow disturbances in damaging endothelial function and altering the existing balance of the endothelial barrier linked to the production of nitric oxide (NO) [82–84]. This damaging activity of the endothelial barrier is also linked to the hyperactivation of the renin–angiotensin–aldosterone system promoted by hyperglycemia since the MAP kinase (MAPK) system is a signal transduction pathway common to insulin and aldosterone and its activation induces proliferation and contractility of vascular myocytes and proinflammatory activity of endothelial cells [85–87]. This mechanism would reduce the production of NO, the most powerful vasodilator in the body and the main indicator of endothelial well-being [81–84,88]. NO plays a crucial role in vessel protection, limiting platelet aggregation and inhibiting the recruitment and adhesion of leukocytes on the damaged vascular surface. In insulin-resistant patients, NO synthesis is compromised and is therefore associated with a greater cardiovascular risk. As IR acts in contributing to endothelial damage, it also contributes to inducing a proinflammatory state and endothelial dysfunction at the level of renal microcirculation and vascular damage, favoring the deposition of matrix by the mesangial cells, inducing glomerulosclerosis [89,90]. As we have already said, before interacting with cellular receptors, LDL can undergo modifications mainly linked to the activity of the cholesterol ester transfer protein (CETP), which mediates the transfer of triglycerides and esterified cholesterol between lipoproteins [90,91]. In particular, triglycerides from VLDL are transferred to LDL and HDL in exchange for cholesterol esters. These interchanges decrease the cholesterol ester content of LDL and increase the triglyceride content, making these particles more susceptible to lipolytic action on the part of hepatic lipase (HL). The end result is the formation of smaller, denser LDLs that are thought to be more atherogenic than normal LDLs. Significant gene–gender interactions have also been reported for the gene that codes for the cholesterol ester transfer protein in its variant TaqIB (CETP) [90,91]. Proatherogenic CETP activity depends on various factors, primarily genetic polymorphisms, but also depends on age, physical activity, alcohol consumption, obesity, and lipid balance, in particular, the concentration of triglycerides, LDL cholesterol, and HDL cholesterol. Villard et al. [92] showed that high-risk cardiovascular patients have

increased CETP activity in comparison to lower-risk patients. In particular, the isoform of CEPT derived from the heterozygous B2 allele was associated with larger particle size for HDL and LDL in men, while in women there was only an increase in HDL particle size. However, the protective association of this genetic variant with cardiovascular risk was present in men but not in women [92]. A polymorphism (I405V) of the CETP gene leading to lower serum C-LDL levels has also been linked to exceptional longevity [93]. The effect of this polymorphism was low for diabetic women who showed significant interactions with the homeostasis model assessment of insulin resistance (HOMA-IR), body mass index (BMI), and concentrations of triglycerides [94,95]. Endothelial damage depends largely on estrogen's influence. In females, estrogen promotes the release of NO and, on the other hand, prevents the production of oxygen free radicals and the chronic inflammatory state deriving from the metabolism of uric acid. Moreover, estrogens play a major role in regulating serum levels of uric acid, resulting in less accumulation in the walls of blood vessels and therefore limiting endothelial damage compared to men. Estrogen activity is the major determinant of gender difference when we associate atherogenesis and endothelial damage and IR, contributing to the damage of the microvessel environment [94,95].

### 7.3. Insulin Resistance and Cognitive and Neurological Diseases

Insulin also has a role in some neuronal functions and is particularly involved in the processes of maintaining the integrity of neurons and increasing the production and sensitivity of some neurotransmitters [96,97]. In general, therefore, it is capable of positively affecting nerve transmission and synapse function. Recently, it has been hypothesized that IR can interfere with the production of some neurotransmitters such as dopamine and is also implicated in a greater deposition of amyloid substances which, associated with a chronic inflammatory state, contribute to the pathogenesis of both Parkinson's disease and Alzheimer's disease [98–100].

In women, the presence of estrogenic activity slows down the progression of Parkinson's disease, as the hormone gives these subjects high levels of physiological dopamine at the nigro-striatal level [101]. This suggests that estrogens may have a role in preventing the disease itself due to their anti-inflammatory activity, which counteracts the chronic inflammatory state induced by IR [102,103]. Overall, therefore, women show a mild course in terms of symptom progression although the clinical severity of symptoms is greater than in their male counterparts [104]. Furthermore, the role of IR in sleep disorders has been hypothesized [105,106]. The discontinuity of sleep has been compared independently from obesity to the syndrome of sleep apnea (OSAS) and as a consequence of the onset of insulin resistance [105–107]. Furthermore, it has been shown that the severity of apneas inversely correlates with insulin sensitivity [108–110] while nocturnal ventilation enhanced by c-PAP improves insulin sensitivity [111]. At the basis of the association between insulin resistance, sleep disturbances, and OSAS is hypoxia which occurs during the apnea phases, increases by the discharge of the sympathetic nervous system, simultaneously inhibits insulin secretion, and reduces the availability of glucose to the arrangement of neurons [111]. Finally, it is hypothesized that sleep disturbances could increase cortisol secretion, and this negatively affects peripheral insulin sensitivity by further amplifying IR [112]. Conflicting data regarding gender exist for the relationship between headaches and IR. In particular, migraine is usually more frequent in women, and in particular, migraine with aura also has an association with metabolic syndrome and cardiovascular risk [113–115]. In two cohort studies, no significant association was observed between migraine and diabetes [114,115]. Studies conducted on animal models have provided a possible link between IR and migraine, considering that several vascular mediators, interleukins or cytokines (TNF-$\alpha$, C-reactive protein, IL-1$\beta$, IL-6, and IL-8), have an important role in both IR and migraine [116,117]. Furthermore, numerous neuropeptides such as substance P, neuropeptide Y, and calcitonin gene related-peptide (CGRP) and some adipokines such as leptin and adiponectin are altered in both conditions [118,119]. On the other hand, population-based and clinical studies have reported that the prevalence of migraine in diabetic patients,

particularly in women, is lower than [120,121], similar to [122], or higher than [123] that in nondiabetic patients. Actually, the presence of diabetes-related vascular changes may potentially increase the risk of migraines or affect the severity of migraines in diabetic women. The use of certain medications for diabetes management may have an impact on migraines. For instance, some medications like metformin and GLP-1 receptor agonists have been reported to potentially reduce the frequency and severity of migraines [124,125]. In migraine-affected women, increased fasting neuropeptide Y levels in migraine may be a factor leading to increased insulin resistance due to specific alterations in energy intake and sympathetic–adrenal system activation [123–126].

### 7.4. Insulin Resistance and Cancer

Recent evidence underlines the association between IR and neoplastic risk, particularly in breast, colorectal, pancreas, and liver cancer [127–132]. Several pathways have been proposed through which IR may have pro-oncogenic, mitogenic, and antiapoptotic effects [133,134]. In fertile women, insulin and IGF-1 inhibit the hepatic synthesis of sex hormone-binding globulin (SHBG), increasing the bioavailability of estrogen, which may account for the increased risk of breast cancer [135,136]. The proinflammatory state and the consequent increased production of free oxygen radicals (ROS) represent a favorable environment for the development and progression of neoplasms inducing mutagenesis and carcinogenesis [137].

### 7.5. Insulin Resistance and Hyperandrogenism

Polycystic ovary syndrome (PCOS) unites hyperandrogenism and IR, occurring in patients who are not exclusively obese but who nonetheless have peripheral IR, albeit variable. PCOS is a condition affecting young patients characterized by marked hyperandrogenism, ovulatory dysfunction, morphological alterations of the ovary, and IR [138]. From the data available in the literature, it has been observed that these conditions are associated with a moderate risk of venous thromboembolism, especially in women with higher estrogen levels [139–141]. The resulting compensatory hyperinsulinemia is caused largely by phosphorylation of the insulin receptor and insulin receptor substrate-1 (IRS-1), which reduces the efficiency of insulin signaling. At the ovarian level, excess insulin stimulates the expression of luteinizing hormone (LH) receptors, making these cells more active and thus promoting the production of androgenic-type sex steroids, leading to anovulation [142,143]. Conversely, all treatments that reduce IR improve ovulation and hyperandrogenism [143].

## 8. Lifestyle Factors

The lifestyle of a man may differ considerably from that of a woman and this makes us understand how the two sexes can contrast in terms of the modifiable risk factors of MS and IR. The data available regarding lifestyle divided by sex show us that women show little propensity for physical activity, with a more sedentary lifestyle which increases with age. On the other hand, they have a healthier diet by consuming fewer foods containing saturated fats and consuming more fiber and vitamins in the form of fruits and vegetables [65]. However, this trend continues if we consider the younger female segments with a more dynamic lifestyle, but also residing in large urban centers, for whom we see an increase in the consumption of unhealthy foods such as street food and an increase in the consumption of sugary and carbonated drinks and therefore an increase in adiposity and all those situations associated with IR [66]. In general, however, if we consider the total mortality in patients who develop conditions related to MS or IR, we note that the percentage of deaths in the female population is only slightly higher than in the male population [67]. In several observational studies, moderate alcohol consumption has been shown to be associated with a lower MS risk. Another meta-analysis based on intervention studies showed that modest consumption tends to improve glycated hemoglobin and insulin sensitivity only in women [68]. This is because it probably has little effect on endogenous estradiol levels without influencing conditions associated with insulin resistance

in postmenopausal women [69]. Surely further efforts are needed to better understand the effect of alcohol on the mechanisms regulating insulin resistance in the two genders. Also, with regard to smoking, there has always been a certain risk for men, traditionally habitual tobacco consumers. In the last twenty years, tobacco consumption has increased in women, particularly young women, who therefore, having more years of exposure, will be exposed to an unpredictable risk of metabolic alterations linked to IR [70], with early mortality from cardiovascular disease also increasing in general in women [71]. The risk of developing metabolic conditions related to IR is also similar in women and men with regard to exposure to secondhand smoke [72].

## 9. Conclusions

Concluding on the relationship between IR and gender medicine involves synthesizing the key findings and insights from existing research. Numerous studies consistently demonstrate that gender differences exist in the prevalence and development of IR. Men and women can exhibit variations in insulin sensitivity, adipose tissue distribution, and hormonal influences, which contribute to these differences. Hormones, particularly sex hormones like estrogen and testosterone, influence insulin sensitivity and resistance. Understanding how fluctuations in these hormones across the lifespan impact IR is critical, especially during puberty, pregnancy, and menopause. The evidence suggests that tailoring treatment approaches based on sex or gender can lead to more effective management of insulin resistance and related conditions. Healthcare providers should consider these differences when prescribing medication and lifestyle interventions. Gender disparities in healthcare access and treatment outcomes related to IR persist. These disparities may result from social, economic, and cultural factors, and efforts should be made to reduce them and ensure equitable care. Gender medicine should move towards a more personalized approach, considering not only biological sex but also gender identity and expression in healthcare. This approach acknowledges that individuals have unique health needs that may not align strictly with traditional binary gender categories. The review may highlight gaps in the existing literature, suggesting areas where further research is needed. Future studies should explore the interplay between genetics, epigenetics, and environmental factors in shaping gender-specific IR. In conclusion, a review of the relationship between IR and gender medicine reveals the complex interplay between biology, hormones, gender identity, and societal factors. Recognizing these factors is crucial for improving healthcare outcomes and reducing disparities in insulin resistance and related conditions. Further research and an inclusive approach to healthcare are essential to address the unique needs of individuals based on their gender and biological sex.

*Take Home Messages*

- IR is a condition in which target cells become less responsive to the hormone, leading to hyperglycemia. IR can affect both men and women, but there are some gender differences in its prevalence, risk factors, and associated health conditions.
- IR is generally more prevalent in men compared to premenopausal women. However, after menopause, the incidence of IR in women increases and becomes more comparable to that of men.
- Differences in body fat distribution between men and women contribute to gender variations in IR. Men tend to accumulate more visceral fat, which is fat stored around the abdominal organs, while women typically have a higher proportion of subcutaneous fat, which is located just beneath the skin. Visceral fat is strongly associated with IR and metabolic disturbances.
- Sex hormones play a role in insulin sensitivity. Estrogen, the primary female sex hormone, appears to have protective effects on insulin sensitivity. Women tend to have better insulin sensitivity during their reproductive years, partly due to the presence of estrogen. However, after menopause, when estrogen levels decline, women may experience a decrease in insulin sensitivity.

- IR is a key underlying factor in the development of T2DM. Men with insulin resistance are more likely to develop T2DM at an earlier age compared to women. Additionally, men with insulin resistance have a higher risk of developing nonalcoholic fatty liver disease (NAFLD) and cardiovascular disease compared to women with insulin resistance.
- Lifestyle choices, such as diet and physical activity, can influence IR. Men and women may have different patterns of dietary preferences and physical activity levels, which can contribute to variations in insulin sensitivity.

**Author Contributions:** Conceptualization, T.C. and P.C.; methodology, G.G.; software, G.G.; validation, T.C., P.C. and M.G.; formal analysis, G.G.; investigation, G.G.; resources, P.C.; data curation, T.C.; writing—original draft preparation, T.C.; writing—review and editing, T.C.; visualization, P.C.; supervision, T.C.; project administration, T.C.; funding acquisition, M.G. All authors have read and agreed to the published version of the manuscript.

**Funding:** This research received no external funding.

**Institutional Review Board Statement:** Not applicable.

**Informed Consent Statement:** Not applicable.

**Data Availability Statement:** Not applicable.

**Conflicts of Interest:** The authors declare no conflict of interest.

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
