# Peer review of "Gender Differences in Insulin Resistance: New Knowledge and Perspectives"

_cimb, doi:10.3390/cimb45100496_

Round 1
Reviewer 1 Report
I read the review titled "GENDER DIFFERENCES IN INSULIN RESISTANCE" by Ciarambino and colleagues.
Some areas for authors to address are as follows:
THE title is narrow however the focus is broad perhaps authors can try to open it up.
keywords are missing.
A presented abstract only detailed a brief background, i still cant find the aim or objectives for the current review. this must be outlined.
One of the limitation of review is that they dont follow specific guide/criteria. perhaps author can specify how they obtained evidence reviewed in this manuscript.
Introduce to topic by providing a detailed background with motivation for conducting this review.
reference are not formatted according to journal guideline. Use square [] in text, I recommend the use of automated reference manager like Mendeley.
line 29: what is syndrome x?
line 35 what is SM?
line 63 author states "... features are summarised in figure 1" however no figure is presented in the manuscript.
line 65 the opening statement need to be revised " as mentioned.."
line 228: reference should be written as 56-59, 63 for consistency with the rest of the manuscript.
two references do not reflect in the manuscript/ not numbered
1. Zou TT, Zhang C, Zhou YF, et al. Lifestyle interventions for patients with nonalcoholic fatty liver disease: a network 706 meta-analysis. Eur J Gastroenterol Hepatol. 2018;30(7):747-755. doi:10.1097/MEG.0000000000001135
2. Munteanu C, Schwartz B. The Effect of Bioactive Aliment Compounds and Micronutrients on Non-Alcoholic Fatty Liver 710 Disease. Antioxidants (Basel). 2023;12(4):903. Published 2023 Apr 10. doi:10.3390/antiox12040903
Author Response
Dear Editor in Chief and Dear Reviewers,
Many thanks for you revision.
We now, send you, our point of point for submission to CIMB
Best regards
REVIEWER 1
Read the review titled "GENDER DIFFERENCES IN INSULIN RESISTANCE" by Ciarambino and colleagues.
Some areas for authors to address are as follows:
THE title is narrow however the focus is broad perhaps authors can try to open it up.
R: According to the reviewer's suggestion, the title has been changed.
keywords are missing.
R: According to the reviewer's suggestion, keywords have been added
A presented abstract only detailed a brief background, i still can't find the aim or objectives for the current review. this must be outlined.
R: According to the reviewer's suggestion, the abstract has been changed, adding aim or objectives.
One of the limitations of reviews is that they don’t follow specific guide/criteria. Perhaps the author can specify how they obtained the evidence reviewed in this manuscript.
R: According to the reviewer's suggestion, a methods section has been added
Introduce to topic by providing a detailed background with motivation for conducting this review.
R: According to the reviewer's suggestion, the background section has been added
reference is not formatted according to journal guidelines. Use square [] in text, I recommend the use of an automated reference manager like Mendeley.
R: According to the reviewer's suggestion, references have been formatted
line 29: what is syndrome x?
R: According to the reviewer's suggestion, the concept of syndrome x has been formatted
line 35 what is SM?
R: According to the reviewer's suggestion, the sentence in line 35 has been corrected
line 63 author states "... features are summarised in figure 1" However no figure is presented in the manuscript.
R: According to the reviewer's suggestion, Figure 1 has been added
line 65 the opening statement needs to be revised " as mentioned.."
R: According to the reviewer's suggestion, Figures have been added
Line 228: Reference should be written as 56-59, 63 for consistency with the rest of the manuscript.
R: According to the reviewer's suggestion, references have been corrected
two references do not reflect in the manuscript/ not numbered
- Zou TT, Zhang C, Zhou YF, et al. Lifestyle interventions for patients with nonalcoholic fatty liver disease: a network 706 meta-analysis. Eur J Gastroenterol Hepatol. 2018;30(7):747-755. doi:10.1097/MEG.0000000000001135
- Munteanu C, Schwartz B. The Effect of Bioactive Aliment Compounds and Micronutrients on Non-Alcoholic Fatty Liver 710 Disease. Antioxidants (Basel). 2023;12(4):903. Published 2023 Apr 10. doi:10.3390/antiox12040903
R: According to the reviewer’s suggestion the two references are removed from the manuscript
Many thanks
Tzziana Ciarambino
MD, PhD
Reviewer 2 Report
The manuscript prepared by Tiziana Ciarambino and Petro Crispino et al. presents a very interesting review of research on gender differences in insulin resistance. Nevertheless, some aspects of the work still need improvement.
Major comments:
1. At the end of the abstract section, please add the purpose of the review.
2. Please add an introduction section with aim of presetned review.
3. Please add methodology sections
4. Please add a chapter on molecular mechanisms: insulin signaling pathway under physiological conditio and insuluin resistance. Please prepare and add one or two figures summarizing and comparing these two states at the molecular level.
5. Please add a summary table of ,,associated health conditions’’.
6. Please add section of Conclusion with summarizing figure.
Minor comments:
1. Please correct the introduction and use of abbreviations throughout the paper.
Author Response
Dear Editor in Chief and Dear Reviewers,
Many thanks for you revision.
We now, send you, our point of point for submission to CIMB
Best regards
REVIEWER 2
The manuscript prepared by Tiziana Ciarambino and Pietro Crispino et al. presents a very interesting review of research on gender differences in insulin resistance. Nevertheless, some aspects of the work still need improvement.
Major comments:
- At the end of the abstract section, please add the purpose of the review.
R: According to the reviewer's suggestion, the abstract has been changed, adding aim or objectives.
- Please add an introduction section with aim of presented review.
R: According to the reviewer's suggestion, the abstract has been changed, adding aim or objectives.
- Please add methodology sections
R: According to the reviewer's suggestion, a methods section has been added
- Please add a chapter on molecular mechanisms: insulin signaling pathway under physiological conditions and insulin resistance. Please prepare and add one or two figures summarizing and comparing these two states at the molecular level.
R: According to the reviewer's suggestion, Figures have been added
- Please add a summary table of associated health conditions’’.
R: According to the reviewer's suggestion, Figures have been added
- Please add a section of the Conclusion with a summarizing figure.
R: According to the reviewer's suggestion, conclusions have been added
Minor comments:
- Please correct the introduction and use of abbreviations throughout the paper.
R: According to the reviewer's suggestion, abbreviations have been corrected
Many thanks
Tzziana Ciarambino
MD, PhD
Round 2
Reviewer 1 Report
Thanks to authors for adequately revising their work, however some minor typographical and syntax errors are noted in a revised version.For instance under method correct 20223 to 2023
Also write the word in full for the first time and abbreviate subsequently, this includes type 2 diabetes mellitus, Insulin resist and metabolic syndrome etc
Author Response
REVIEWER 2
Comments and Suggestions for Authors
Thanks to the authors for adequately revising their work, however, some minor typographical and syntax errors are noted in a revised version. For instance, under method correct 20223 to 2023
R: According to the suggestion of the reviewer the sentence has been corrected
Also, write the word in full for the first time and abbreviate subsequently, this includes type 2 diabetes mellitus, Insulin resistance and metabolic syndrome etc
R: According to the suggestion of the reviewer the abbreviation has been completely revised
REVIEWER 2
Comments and Suggestions for Authors
Thanks to the authors for adequately revising their work, however, some minor typographical and syntax errors are noted in a revised version. For instance, under method correct 20223 to 2023
R: According to the suggestion of the reviewer the sentence has been corrected
Also, write the word in full for the first time and abbreviate subsequently, this includes type 2 diabetes mellitus, Insulin resistance and metabolic syndrome etc
R: According to the suggestion of the reviewer the abbreviation has been completely revised
REVIEWER 2
Comments and Suggestions for Authors
Thanks to the authors for adequately revising their work, however, some minor typographical and syntax errors are noted in a revised version. For instance, under method correct 20223 to 2023
R: According to the suggestion of the reviewer the sentence has been corrected
Also, write the word in full for the first time and abbreviate subsequently, this includes type 2 diabetes mellitus, Insulin resistance and metabolic syndrome etc
R: According to the suggestion of the reviewer the abbreviation has been completely revised
REVIEWER 2
Comments and Suggestions for Authors
Thanks to the authors for adequately revising their work, however, some minor typographical and syntax errors are noted in a revised version. For instance, under method correct 20223 to 2023
R: According to the suggestion of the reviewer the sentence has been corrected
Also, write the word in full for the first time and abbreviate subsequently, this includes type 2 diabetes mellitus, Insulin resistance and metabolic syndrome etc
R: According to the suggestion of the reviewer the abbreviation has been completely revised
Reviewer 2 Report
The authors did not address all my comments. I am asking you again to refer and make the following changes:
Major comments:
4. Please add a chapter on molecular mechanisms: insulin signaling pathway under physiological condition and insuluin resistance. Please prepare and add one or two figures summarizing and comparing these two states at the molecular level.
5. Please add a summary table of ,,associated health conditions’’.
Author Response
REVIEWER 3
Comments and Suggestions for Authors
The authors did not address all my comments. I am asking you again to refer and make the following changes:
Major comments:
- Please add a chapter on molecular mechanisms: insulin signaling pathway under physiological conditions and insulin resistance. Please prepare and add one or two figures summarizing and comparing these two states at the molecular level.
R: According to the suggestion of the reviewer a new section regarding insulin signalling pathway has been added
- Please add a summary table of associated health conditions’’.
R: According to the suggestion of the reviewer a new table regarding “associated health conditions has been added
REVIEWER
ReComments and Suggestions for Authors
The authors did not address all my comments. I am asking you again to refer and make the following changes:
Major comments:
- Please add a chapter on molecular mechanisms: insulin signaling pathway under physiological conditions and insulin resistance. Please prepare and add one or two figures summarizing and comparing these two states at the molecular level.
R: According to the suggestion of the reviewer a new section regarding insulin signalling pathway has been added
- Please add a summary table of associated health conditions’’.
R: According to the suggestion of the reviewer a new table regarding “associated health conditions has been added
Round 3
Reviewer 2 Report
I accept manuscript in present form.